



# Quantifying the impact of land cover changes on hydrological extremes in India

Shaini Naha*, Miguel A Rico-Ramirez and Rafael Rosolem
Civil Engineering Department, University of Bristol, Bristol, BS8 1TR, UK

5          * Corresponding author email: sn17546@bristol.ac.uk

## Abstract

Several research studies have addressed the effects of future climate changes on the hydrological regime of Mahanadi river basin located in eastern part of India. However, studies

investigating the effects of future land cover changes on hydrology are limited owing to the lack of availability of projected land cover scenarios. Our study investigates how the hydrology of Mahanadi river basin would respond to the current and future land cover scenarios under a large-scale hydrological modelling framework. Both historical and future land cover scenarios from the recently released, Land use Harmonisation (LUH2) project for CMIP6,

indicates cropland and forest are the major land cover types in the basin with a noticeable increase in the cropland (23.3%) at the expense of forest (22.65%) by the end of year 2100 compared to the baseline year, 2005. A physically semi-distributed model, the Variable Infiltration Capacity has been set up and implemented over the Mahanadi river basin system for the time period 1990-2010. The uncertain model parameters were subjected to Sensitivity

Analysis and calibrated within a Monte Carlo framework. The best set of calibrated models obtained is used in conjunction with the harmonized set of present and future land use scenarios from LUH2 at 25 km by 25 km resolution to generate an ensemble of model simulations that captures a range of plausible impacts of land cover changes on discharge and other hydrological components of the basin. Overall, model simulation results indicate an

increase in the extreme flows (i.e., 95th percentile or higher) in the range of 0.12 to 21 % at multiple subcatchments within the basin. This increase can be attributed to the direct conversion of forested areas to agriculture (on the order of 30,000 km$^2$) that has reduced the Leaf Area Index and subsequently reduces the Evapotranspiration (ET). These changes ultimately affect other water balance components at the land surface, resulting in an increase

in surface runoff and baseflow, respectively.


**Keywords:** Land cover change, Variable Infiltration Capacity (VIC), LUH2, Sensitivity Analysis, calibration, hydrological components

## 1. Context and Background

Land use and land cover change (LULC) induced by the rapid anthropogenic activities, is one of the major causes of change in hydrological and watershed processes (Rogger et al., 2016). Alterations of existing land cover types and land management practices in a catchment can thereby, significantly modify the rainfall path into runoff by changing the hydrological dynamics such as surface runoff, baseflow, Evapotranspiration (ET), water holding capacity of

the soil, interception and groundwater recharge , thus reflecting a change in the water demand (Berihun et al., 2019; Bosch and Hewlett, 1982; Costa et al., 2003; Foley et al., 2005; Garg et al., 2017; Hamman et al., 2018; Mao and Cherkauer, 2009; Zhang et al., 2014). Rapid growth in population in the developing countries has prominent effects on LULC dynamics through deforestation at the expense of increased agricultural production. Deforestation

among all other land use changes is the major cause of modifying various hydrological processes such as ET, surface runoff, baseflow and snowmelt processes (Dwarakish and Ganasri, 2015; Gao et al., 2009). The complex relationships between the human induced land cover change and the hydrological processes have gained widespread attention among various scientific communities across the world. In this regard, several studies have been

carried out that links the LULC changes and the hydrological dynamics within a river basin (Abe et al., 2018; Behera et al., 2018; Berihun et al., 2019; Chu et al., 2010; Costa et al., 2003; Rogger et al., 2016; Thomson et al., 2018; Wang et al., 2008; Wilk and Hughes, 2002). However, the exact role of LULC changes in modifying river discharge is still elusive (Rogger et al., 2016) and therefore, remains a challenge to isolate the sole impacts of land use changes

on hydrology of a river basin (Tsarouchi and Buytaert, 2018). The challenge also lies in solving these complex processes in a heterogenous catchment coupled with limited hydro-climatological data (Gebremicael et al., 2019; Li and Sivapalan, 2011).

Changes in land use and cropping patterns are modifying the hydrological cycle in many river basins of India. In 1980's, Central, North-Eastern and Peninsular India was bestowed with

woody savannas which is mostly forest lands (Paul et al., 2016). Since then, rapid urbanization and agricultural intensification are the constant reasons behind the depletion of natural





vegetations and conversion of woody savannas to the cropland. As per the Land Use and Land

Cover (LULC) map of 2005, cropland is the dominant land cover type in India. The analysis

report of world "greening" from MODIS (2000-2017) showed a significant increase of 82% in

the greening over India, which is found to be entirely associated with cropland (Chen et al.,

2019; IPCC, 2019). Several researchers (Babar and Ramesh, 2015; Bosch and Hewlett, 1982;

Gebremicael et al., 2019; Wilk and Hughes, 2002) agree that the expansion in agricultural land

at the expense of vegetative cover results in an increase in surface runoff and decreases river

discharge in a given watershed. Wilk and Hughes, (2002) found a maximum increase of 19%

in the runoff from a tropical catchment in South India due to the expansion in agriculture at

the expense of forest and savannas. Babar and Ramesh, (2015) estimated an increase in

runoff (0.9%) and decrease in ET (4.5%) due to the conversion of forest to agriculture and

built up areas in Ganga basin in India. The population of India is expected to increase by an

average of 36% in the 2050's and by 108% at 2090's thereby rendering changes to agriculture

and water demand (Jin et al., 2018). Many river basins of India that have undergone drastic

land cover transformation over the years have been facing extreme hydrological events like

floods and droughts in recent times. Long-term changes in climate and land use are reported

as the main reasons causing these hydrological extremes (IPCC, 2019).

Climate and LULC governs the hydrological cycle in a basin through an intricate relationship

involving a wide range of interactions among the land surface variables at different spatial

and temporal scales. These interactions can be best solved through the implementation of

process-based and physically based distributed or semi-distributed hydrological models,

representing the land surface characteristics of a heterogeneous catchment, and simulating

the multi-layered hydrological processes. Therefore, the selection of an appropriate

hydrological model is quite essential. The Variable Infiltration Capacity (VIC) model is a large

scale physically semi-distributed land surface model developed by Liang et al., (1994). The

ability of the model to simulate the impacts of LULC changes on hydrology are well

documented in various research articles (Garg et al., 2017, 2019; Hurkmans et al., 2009; Mao

and Cherkauer, 2009; Patidar and Behera, 2019; Zhang et al., 2014).

Eastern part of India is amongst the most rapidly changing landscape over the country,

specifically, Mahanadi river basin in Eastern India have undergone drastic land cover changes

in the last decades (Behera et al., 2018; Dadhwal et al., 2010). To the best of our knowledge,

only one study, Das et al., (2018) predicted the land cover change impacts on the future (year



2025) water balance components of Eastern Indian river basins. Dadhwal et al., (2010)

simulated the effects of deforestation and agricultural expansion on the annual streamflow of Mahanadi river basin from 1972-2003. Nonetheless, these studies are based on manually calibrated singular model experiment and therefore lacked the influence of model parameter uncertainties on the outcome. Single prediction obtained from a calibrated model can be biased whereas an analysis based on ensemble of calibrated models provides a range of

possible hydrological changes which is more robust and reliable, hence, effective in aiding decision in context of water resources management. Several recent studies ensured the need of considering model parameter uncertainties while modelling the changes in hydrology which shall have implications on water resource management, allocation and supply. Chaney et al., (2015) used an ensemble of behavioural parameter sets while monitoring global flood

and drought to provide predictions along with the uncertainty estimates. Singh et al., (2014) identified the impact of combined climate and land use projections on hydrology while also considering different model parameter sets and uncertainties associated with it in the analysis. Zhang et al., (2019) used best performing model parameter sets and found variability in the projected hydrological variables owing to the parameter uncertainties involved.

Furthermore, the LULC studies carried out in Eastern India or Mahanadi river basin have used aggregated (monthly) time steps in modelling the change, that miss the dynamics of daily flow variability.

In this study, we address the overall science question: What is the isolated role of LULC change on the water balance of a large river basin? To understand this, two specific research

questions arises are: (1) Can we identify changes in surface water balance due to changes in land cover while using a regional hydrological model? and (2) What are the uncertainties associated with model parameters obtained for the regional consequences and how they affect simulated water balance components? This paper specifically focusses on the Mahanadi river basin in India. We identify best ensembles of daily model simulations

calibrated within a Monte Carlo Framework, which accounts for the model parameter uncertainties, to evaluate the LULC changes impacts on the hydrology of the basin. The land cover scenarios used in this study represents future changes in the LULC under changing climate (RCP's) and socio-economic conditions (SSP's). This is the first study that uses applications of VIC model in conjunction with gridded land cover forcing's under combined

SSP and RCP scenarios in Mahanadi river basin. The outcome of this study is a range of



hydrologic predictions associated with the model parameter uncertainties owing to the land cover changes occurring in the Mahanadi river basin, allowing better understanding and implementation of the adaptation and mitigation strategies in the future.

## 2.  Research Area

Geographical Overview

The Mahanadi river basin is located in the eastern part of India (Figure 1) and drains an area of 141,589 km$^2$, which nearly accounts for 4.3% of the total geographical area of India. The

basin has a varying topography with its lowest elevated area (-17 m) lying in the coastal reaches and the highest elevated area (1323 m) in the northern hills. The basin is characterized by tropical climate zone and receives rainfall from southwest monsoons which commences in June and lasts till September. The average annual rainfall is 1572 mm, with ~ 78% of the total annual rainfall occurring during the monsoon months. The basin is also

subjected to spatial variability in terms of receiving rainfall which has resulted in floods in some parts of the basin and drought in others. The mean annual discharge is 1895 m$^3$/s. Hirakud dam with a gross storage capacity of 8.136 km$^3$ is the major hydro project in the river basin constructed in the year 1957 to alleviate the flood problems and to serve multiple other purposes such as irrigation, hydropower generation and supplying drinking water. About 65%

of the basin is placed upstream of the dam. Despite its significant storage capacity, the large flows from the catchment upstream as well as from the middle reaches (i.e. between Hirakud dam and Mundali weir) causes devastating floods during the monsoon in the deltaic region of the basin.

About 48% of the total area is under agriculture (Figure 2a), out of which 30% is cropped

during the kharif season (monsoon season) and 15% is under double or triple irrigation. The remaining 3% of the area is either cropped during Rabi season (Spring) or Zaid (summer) season. Deciduous Broadleaved Forest (DBF) being dominant among other forest types, covers 25% of the basin area (Figure 2a). Built up, plantation, grassland, scrubland, water bodies and other forest types constitute the rest 22 % of the basin area. Comparison of the

LULC maps of 2005-2006 and 2014-2015 derived from National Remote Sensing Centre (NRSC) shows a substantial increase in the agricultural land from about 42.5 to 48% at the expense of fallow land, built up areas and water bodies. Forest cover almost remained same





with a minimal increase of only 0.3% (NRSC, 2014). In addition, loamy and clayey are the major soil types covering 53.33% and 41.5% respectively of the total basin area (NBBSS-LUP, India).

Approximately 90% of the basin has moderately shallow to deep soil having depth greater than 50 cm.

## 3. Materials and Methods

### 3.1 Model structure and Implementation

The VIC-3L model is a semi-distributed macroscale hydrological model which solves either only water balance or full water and energy balance at each grid cell for three soil layers (Cherkauer and Lettenmaier, 1999). The model in water balance mode assumes air

temperature to be same as the surface temperature. The most distinguishable features of this model includes, maintaining sub-grid heterogeneity for the vegetation covers and sub-grid variability in soil moisture storage capacity (Liang et al., 1994), causing surface flow considering both infiltration excess and saturation excess (Bao et al., 2011) and occurrence of baseflow from the third soil moisture layer as a non-recession flow (Zhao et al., 1980).

In VIC-3L, direct runoff occurs from the top thin layer. The Middle soil layer allows for diffusion of water to the uppermost soil layer provided the middle soil layer is wetter. Evaporation occurs from all the soil layers and baseflow occurs from the third layer. Sub grid spatial variability in soil moisture storage is represented by a variable infiltration curve where the model assumes that the infiltration capacity is the non-linear function of the soil moisture

storage within the grid cell (Liang et al., 1994). More details regarding the structure and formulations of the model can be found in the literature (Gao et al., 2010; Liang et al., 1994). To obtain the discharge at the outlets of multiple subcatchments, the VIC-3L model is coupled to a stand-alone routing model (Lohmann et al., 1996). Lohmann's routing model follows a simple river routing scheme where runoff and baseflow are first routed to the edge of the

grid cells using an instantaneous unit hydrograph and finally transported to the river/channel network using a linearized St. Venant's equation.

The VIC model has been widely applied at multiple scales ranging from global to continental to large scale basins under various land use/climate scenarios (Dadhwal et al., 2010; Hurkmans et al., 2009; Mao and Cherkauer, 2009; Matheussen et al., 2002; Mishra et al.,

2010; Yang et al., 2014). Also the model has been used successfully in simulating hydrological





processes in many Indian river basins under different environmental conditions (Garg et al., 2017, 2019; Mishra et al., 2008, 2010; Naha et al., 2016; Patidar and Behera, 2019). Patidar & Behera, (2019) examined the effects of conversion of natural vegetation to urban areas and agriculture on the VIC simulated hydrological fluxes in the Ganga river basin in India. Garg et

al., (2019) successfully implemented the VIC model to simulate the fluxes in Pennar river basin in India under changing land cover conditions.

In this study, the VIC-3L model has been implemented over multiple subcatchments of the Mahanadi river basin system using 3 root zones to evaluate the hydrological responses to the future LULC changes at a grid size of approximately 5 km. The model has been run in water

balance mode. The subcatchments analysed are Basantpur (Ba), Kantamal (Ka), Kesinga (Ke), Salebhata (Sa) and Sundergarh (Su) (Figure 1). The input data required to drive the model and its sources are described in the subsequent section.

### 3.2 Model setup

A wide range of temporal and spatially distributed datasets such as meteorological forcing, soil and land cover information and topographic features are required to drive a physically distributed hydrological model. Topographical features are determined using the 30-meter CARTO-DEM (Cartosat-1 Digital Elevation Model), a national DEM developed by ISRO (Indian Space Research Organization) (Sivasena Reddy and Janga Reddy, 2015). The delineated river

basin is converted into grid format of resolution 0.05 degrees constituting of 4807 grids within the basin area. Soil textures are derived from the digitized soil map as provided by National Bureau of Soil Survey and Land Use Planning (NBBSSLUP) (Scale 1:250000). Specific soil properties (See Table 1) such as initial soil moisture content, Fractional soil moisture content at critical point, Wcr_frac and Fractional soil moisture content at wilting point, Wp_frac are

calculated based on average hydraulic properties of USDA soil textural classes (Cosby et al., 1984; Rawls et al., 1998; Reynolds et al., 2000). The LULC map which is used in the model runs while performing sensitivity analysis, model calibration and validation is derived from NRSC of year 2005 (scale 1:250000; resolution 56 meters) which was reformatted and reclassified into USGC LULC types as required by the VIC model (Figure 2a). The root zone depth of each

LULC  types and the fraction of vegetation roots in each root zone is obtained from the literatures of (Nijssen et al., 1997; Raje et al., 2014; Zeng, 2002). Other vegetation properties in the vegetation library file such as LAI, roughness length, albedo, architectural resistance,

stomatal resistance and displacement height are assembled based on Global Land Data
Assimilation System (GLDAS). LAI is known to exert strong influence on runoff and ET

simulated by the VIC model (Gao et al., 2010; Matheussen et al., 2002). Hence, we derived
the daily LAI product (2000-2015) from MODIS AQUA/TERRA and compared with the LAI
values from GLDAS database for the river basin. Monthly mean LAI of all the LULC types from
MODIS captures the phenological characteristics better than the GLDAS LAI (Fig 2b) which
shall have further implications on water balance. The range of MODIS LAI for each LULC type

are well in agreement with the LAI values obtained by Patidar and Behera, (2019) for Ganga
river basin in India. The primary meteorological inputs required to drive the VIC-3L model are
precipitation, max. temperature, min. temperature and wind speed. Daily gridded
precipitation (0.25deg by 0.25deg) and daily gridded maximum and minimum temperature
(1deg by 1deg) for the time period (1988-2010) are obtained from India Meteorological

Department (IMD) (Pai et al., 2014). The observed discharge data for calibration and
validation at multiple gauges (Figure 1) are available from Central Water Commission (CWC),
India from the same time period.

### 3.3 Model calibration and validation

VIC-3L model parameters are at first subjected to Sensitivity Analysis (SA) in priory to define
the key parameters needed to be calibrated. Table 1 lists all the important VIC-3L model
parameters that are either estimated or subjected to SA and model calibration. The uncertain
parameters to be included in the SA process and their ranges (Table 1) are set according to
various published literatures both in global and Indian context as shown in Table 1 (Demaria

et al., 2007; Matheussen et al., 2002; Mishra et al., 2008; Park and Markus, 2014; Shwetha
and Varija, 2015; Troy et al., 2008; Xie et al., 2007) and some initial model experiments. The
soil parameters Exp and Ksat were assumed to be the same for all three soil layers.
Sensitivity Analysis is performed explicitly for all subcatchments of Mahanadi river basin
(Figure 1) using a Global Sensitivity Analysis (GSA) technique, Elementary Effect Test (EET)

(Morris, 1991) and three objective functions: Nash-Sutcliffe Efficiency (NSE), Log
transformation of NSE (lnNSE) and Klein-Gupta Efficiency (KGE) were included in the analysis.
NSE focuses on high flows (Nash and Sutcliffe, 1970) whereas lnNSE focuses on low flows. KGE
is the improved version of NSE, gives equal weight to the high and low flows (Gupta et al.,


2009). Ranking of the uncertain model parameters have been generated according to their

relative contribution to the output variability by analyzing the sensitivity indices (Pianosi et al., 2016). Parameters which showed poor performance when tested across all the subcatchments and objective functions were discarded.

The VIC model calibration is performed semi-automatically using a sequence of Monte-Carlo simulation where 10000 near-random parameter sets (influential parameters) were

generated from within the specified range using LHSM with uniform distribution. We abstained from calibrating and validating the model for the entire Mahanadi river basin due to the presence of a major water management structure, Hirakud dam at the middle reach of the basin. Instead the model was run, calibrated, and validated daily for each parameter set for the time period (1990-2000) at all the subcatchments Basantpur, Kantamal, Kesinga,

Salebhata and Sundergarh following 2 years of warm-up period (1988-1989). Hence, calibrated model parameters vary from one subcatchment to another. Next, a pareto set of solutions (parameters) (Bastidas et al., 1999; Efstratiadis and Koutsoyiannis, 2010) are generated for all the subcatchments according to the various trade-offs among different catchment characteristics through the maximization of NSE. Therefore, to obtain a single

parameter set for the entire basin, a pareto rank has been assigned to each of the 10,000 model parameters in the pareto solution. Model simulations which have obtained 'Rank 1' in the Pareto Ranking analysis are considered as the 'best' performing simulations, which are further used to generate the streamflow series in the validation period (2001-2010) for all the subcatchments. The ensemble of calibrated models is then used to assess the impacts of land

cover changes on the hydrology of Mahanadi river basin.

### 3.4 Historical and Future Land use land cover scenarios

All the model simulations in the calibration and validation period (Section 3.3) are obtained using a static local LULC map of year 2005 derived from NRSC. Simulations using this land use map shall be termed as NRSC2005 henceforth.

Next, we used a harmonized set of land use scenarios which is a  combination of the Socio-economic Pathways (SSPs) and Representative Concentration Pathways (RCPs) from the recently released , Land Use Harmonization Project (LUH2) (release "LUH2v2h" and LUH2v2f) for the time period of (850–2005) and (2015-2100) respectively (Hurtt et al., 2018)  (Table S1



in the Supplementary section). The LUH2 approach estimates the gridded land use fractions,

annually at a resolution of 0.25 deg.

The LULC of year 2005 from LUH2 is processed and converted to a LULC map of Mahanadi river basin extent showing a single vegetation coverage at each grid size of 0.25 deg. It is then converted to the VIC model grid size of 0.05 deg. The land use classes are reduced in order to simplify our model application, and consequently remapped to the VIC land use classes by

assuming all primary (Forested or Non-forested) and secondary (Forested and Non-forested) land to Deciduous Broadleaf Forest (DBF), Managed pasture and Rangeland are considered as Grassland and all crops are merged into a single Cropland class. Urban land and water bodies are retained (Table S2 in the Supplementary section). It is worth mentioning that the 'potentially non-forested secondary land' class in the LUH2 datasets matched to the forested

areas in the LULC map from NRSC and hence they are both mapped into (Deciduous Broadleaf Forest) DBF which is the dominant forest type in the basin (Fig S1 in the Supplementary section). Simulations using this land use map shall be termed as LUH2005 henceforth.

Spatial LULC maps, NRSC2005 and LUH2005 have been compared (Figure 3) prior to the model simulations to have more confidence in the future scenario and to be able to use LUH2005 as

the baseline scenario. Spatial patterns of Cropland (CL) and Forest (F) which are the most dominant land-use classes in the area shows a similar spatial trend when inspected visually (Figure 3). Also, the percentage of area covered by each land use classes are shown in Table 2. The only notable difference is the lack of Barren ground (BG) in LUH2005 compared to about 12% coverage in the NRSC2005 database. However, the areal coverage of the two most

important classes in the river basin (DBF and CL) show highly comparable percent values between the two products. Note that we will refer to DBF as Forest (F) henceforth. Finally, the VIC model is run for the validation period (2001-2010) using both land use maps from LUH2005 and NRSC2005 and the model set up with LUH2005 was used as a baseline scenario for analysing impacts of land cover changes in the future.

The fraction of basin area occupied by land use classes were computed for all the scenarios of the LUH2 dataset. However, due to the large computational demand of our simulations, we only considered the 'worst' case scenario, RCP3.4 SSP4, which resulted in maximum change in the fractional area of the land use types in the future years (Figure 4). Land cover changes and fractional area covered in other scenarios are shown in Fig S2 in supplementary

section. Four distinct years have been chosen for this study: 2005 (Historic), 2015 (Present),

2050 (Near Future) and 2100 (Far Future) to study the impacts of LULC change in the Mahanadi river basin. A sharp decrease in the forest cover is observed at the expense of agriculture in the years 2050 and 2100 (Figure 4).

The ensemble of calibrated models have been run thrice using the LUH datasets: (1) with land

use map 'LUH2015' as an input which is termed as the 'present' (P) scenario (2) with land use map 'LUH2050' as an input which is termed as the 'Near Future' (NF) scenario (3) with land use map 'LUH2100' as an input which is termed as the 'Far Future' (FF) scenario. To account for the extreme hydrological effects that these changes could cause, three hypothetical scenarios are framed and the models are also run for (1) 'All Cropland' (CL) scenario where all

the grassland and forest areas are transformed into cropland (2) 'All Forest' (F) scenario where all the cropland and grassland areas are transformed into forest (3) 'All Grassland' (GL) scenarios where all the cropland and forest areas are transformed into grassland. The urban and water bodies in these hypothetical scenarios are retained as per the baseline scenario. In all the six cases of model run, meteorological forcing is held constant i.e. the daily

precipitation, maximum and minimum temperatures and wind speed are to be kept same for all the scenarios. Therefore, any changes to be observed in the hydrological components in the above-mentioned scenarios will be due to the sole effects of land cover change. The percent areas covered by each land use classes within the entire basin in year 2005, 2015, 2050 ,2100 and the hypothetical scenarios are shown in Table 3. The percent areas covered

by each land use classes in all the subcatchments are shown in Table S3 in the supplementary section.

## 4.  RESULTS

### 4.1 Model Calibration and Validation

Sensitivity Indices, computed as mean and standard deviation, of the Elementary Effect Test (EET) are computed at the parameter identification stage of model calibration to identify the influential and non-influential parameters. Figure 5 shows the sensitivity of VIC model parameters on river flows represented by the Sensitivity Indices (normalised Mean) of EET method across all the subcatchments and objective functions. Parameters such as

unsaturated hydraulic conductivity, Exp, and maximum velocity of baseflow, Dsmax, are the most important parameters across all the subcatchments and objective functions (NSE, KGE and lnNSE) which indicates that these parameters are sensitive to both high and low flows.



However, sensitivity of these parameters varies among the subcatchments and also within a subcatchment depending on the objective functions used. For example, the most influential

parameters for Basantpur (largest subcatchment) based on objective function, NSE, are the depth of second and third layer of soil, d2 and d3, and in case of Sundergarh (smallest catchment) are Exp and infiltration parameter, binf. Furthermore, the most influential parameters for Basantpur based on NSE, which focusses on high flows are d3 and d2 whereas based on lnNSE, which focusses on low flows, are Exp and Dsmax. in overall, the parameters

sensitive to lnNSE are mainly the baseflow   parameters (Exp, Dsmax, ds, Ws and d3) and are slightly different from NSE and KGE (d2, exp, Dsmax, binf, vel). The depth of first soil layer, d1, saturated hydraulic conductivity, ksat and routing parameter, diff are non-influential for all the subcatchments based on all the objective functions. Thus, these three parameters are eliminated from the calibration experiments and set to specific values based on literatures

and some model experiments.

After eliminating three non-influential parameters, eight (out of eleven) model parameters are subjected to model calibration for the time period (1990-2000). Figure 6(a) shows the range of NSE values computed on a daily scale for 10,000 simulations, calibrated simulations (around 100 simulations obtained with 'rank 1' in the Pareto ranking analysis) and validated

simulations for all the subcatchments in the highest order of their catchment size. The uncertainties and the outliers in the flow values in both calibration and validation period is considerably less compared to that of 10,000 simulations with an improved NSE range at all the subcatchments. The average NSE values in the calibration period at Basantpur, Kantamal, Kesinga, Salebhata and Sundergarh are 0.73, 0.80, 0.70, 0.62, 0.5 respectively and in the

validation period are 0.68, 0.76, 0.69, 0.55 and 0.55 respectively. The performance of the ensemble models in simulating daily flows at Basantpur (Ba), Kantamal (Ka) and Kesinga (Ke) are mostly within the "good" and "very good" range of NSE in both calibration and validation period according to Moriasi et al., (2007) with the model performing slightly better in the calibration period. The NSE values lying within the "satisfactory" range for Salebhata is the

poorest performing station among all subcatchments. The range of NSE'S for the daily calibration and validation at all subcatchments are listed in Table S4 in Supplementary section. The models were able to simulate the daily flows consistently when compared to the observed flows and reproduce the peak flows at different subcatchments in both calibration





and validation period (Fig S3 in Supplementary section). The grey lines in the parallel

coordinate plot in Figure 6b shows the distribution of input parameters within their variability

range and the lines highlighted in black are the parameter sets that have resulted in overall

good model performance across the entire Mahanadi river basin obtained through Pareto

ranking.

### 4.2 Control case scenario performance

We use the calibrated VIC models with LULC maps from two distinct sources, global LUH2005

and regional NRSC2005, which are configured to the model grid resolution. Model

performance of LUH2005 in comparison to NRSC2005 helps to evaluate the robustness of the

future LUH scenarios prior performing the model simulations using future LUH projections.

The Boxplot in Figure 7 shows the range of NSE for daily simulations, LUH2005 and NRSC2005

for all the subcatchments arranged in decreasing order of their performance with regard to

NRSC2005. For NRSC2005, the median NSE values computed for the sub basins Ba, Ka, Ke, Su

and Sa are 0.68, 0.76, 0.69, 0.55 and 0.55 respectively and for LUH2005 are 0.63,0.68, 0.61,

0.56 and 0.52 respectively, showing good overall agreement. The median NSE in LUH2005

suggests that the best performance is at Kantamal followed by Basantpur and Kesinga and

relatively poorer performance at Salebhata and Sundergarh respectively. These trends in the

NSE values are well in agreement with NRSC2005 where the best median NSE is obtained for

Kantamal followed by Kesinga and Basantpur and relatively poor performance at Salebhata

and Sundergarh. There is a systematic reduction in NSE values using LUH2005 at all the

subcatchments barring the smallest subcatchment, Salebhata which has shown a consistent

performance using both the local and global LULC maps. The slight performance drop in the

baseline scenario using LUH2005 against NRSC2005 is due to the average tendency of the

model to underestimate the simulated flows. (see percent bias Fig S4 in Supplementary

section). Land cover class, Barren Ground is non-existent in LUH2005 unlike NRSC2005 and

has been replaced by CL (4%), F (5.02%), GL (4.57). The increase in flows due to the increase

in cropland might have been compensated by the decrease in flows due to the increase in

forest. Therefore, the underestimation in the simulated flows using LUH2005 may result from

the increasing grasslands which increased LAI, thus resulting in an increase in ET and decrease

in surface runoff respectively. The difference in resolution between land cover maps,

LUH2005 and NRSC2005 has led to these differences in the aerial coverage of the land cover

types. However, in overall, objective function, NSE suggests both distinct land cover maps for the baseline scenario from NRSC and LUH show comparable model performance in the historical period with the model being able to capture the seasonality and Land Use/ Land Cover dynamics while simulating the daily flows.

### 4.3 Impact of land use changes

Figure 8 shows percent change in annual average of extreme flows (i.e., 95th percentile or higher) in the Near Future (NF), Far Future (FF), and 3 hypothetical scenarios with respect to baseline scenario LUH2005 for the ensemble of calibrated models. The subcatchments in the boxplots are arranged in the highest order of their performance in the baseline scenario. Percent change in mean annual flows in the future and hypothetical scenarios are also

analysed and shown in Fig S5 in supplementary section.

Projected streamflow for the time period (1990-2010) indicates almost no change to 1% change in extreme flows in the present (P) scenario i.e. model simulations using land use map LUH2015, hence not shown Figure 8. The marginal increase in cropland by 1.4% in the P scenario that would have led to an increase in the flows, however, have been compensated

by increase in forest by 1.5%. Increase in forested areas tends to increase ET with the help of increased LAI, thereby reducing the flows. Also, the comparison of regional land use map of 2005 and 2014 from NRSC (not shown here) shows a negligible change in the forested area. But there is an increase in the agricultural area (7%) at the cost of fallow land (7%). However, this change is not observed in LUH2015 because of the inexistence of the class 'fallow land'

which has been replaced by the cropland, forest, and grassland.

The effect of LULC change on the hydrological fluxes are prominent in the 'Mid Future', 'Far Future' and the hypothetical scenarios. An overall increase in the annual average of extreme flows of (2-16) % for NF and (2-20) % in FF scenario are observed across the Mahanadi river basin and the entire set of models. The median percent change in the NF and FF scenarios at

all the subcatchments lies within the range of (2-5) %. Percent change of slightly higher magnitudes are noticed in the mean annual flows in the NF (1.6-17.4) % and FF (1.5-21.7) % scenarios (Table S5 in Supplementary section). The increase in the annual average of extreme flows in NF and FF scenarios can be attributed to the overall reduction in forest cover by 15.55% and 22.65% and an increment of cropland 13.65% and 23.3% respectively. However,

changes in land use area varies from one subcatchment to another (Table S3). Maximum



increment in cropland (37%) at the expense of forest (38%) is observed at Basantpur and the minimum increment in cropland (16%) at the expense of forest (14%) is observed at Salebhata in the FF scenario.

Percent increase of (0-32) % is observed in 'All Cropland' (CL) scenario and percent change in
the 'All Forest' (F) scenario ranges from an increase of 0.5% to a decrease of 40%. Percent change in the Grassland (GL) scenario ranges from an increase of 12% to a decrease of 3%. The median percent change in CL scenario are slightly higher than that of FF scenario whereas the median values suggest negligible change in the GL scenario. The median percent change in F varies from -3% to -12% across the subcatchments. Maximum increase and decrease in
percent change are observed in CL and F scenario respectively. In GL scenario, percent decrease in the flows are less than the F scenario whereas percent increase is less than the CL scenario. A maximum increase in the annual extreme flows of 830 cumecs is noticed in CL scenario followed by an increase of 532 cumecs in the FF scenario at Basantpur (Table S5 in Supplementary section).

Because the changes in the extreme and mean flows are of almost similar magnitudes at all the subcatchments, other water balance components are analysed with respect to the entire basin to understand the factors causing changes in the streamflow in overall. Figure 9 shows the percent differences in the water balance components in the NF, FF and the hypothetical scenarios, CL, F and GL. Percent change in the P scenario is negligible, hence not shown in the
boxplot. NF scenario depicts an increase in the surface runoff, baseflow and soil moisture content within a range of (1.5 to 9) %, (3 to 26) % and (2 to 7) % respectively followed by decrease in ET within a range of – (1.6 to 3.3) %. FF scenario depicts an increase in the surface runoff, baseflow and soil moisture within a range of (1.5 to 12) %, (4.9 to 32) % and (2.2 to 10) % respectively followed by decrease in ET within a range of -(1.8 to 3.5) %. The median
percent change in runoff, ET, baseflow and soil moisture content in the FF scenario are 4%, -2.2%, 13% and -4% respectively. Percent change in the CL scenario depicts an increase in the surface runoff, baseflow and soil moisture content within a range of (1 to 20) %, (2 to 50) % and (2.2 to 16) % respectively followed by decrease in ET within a range of (0.5 to7) %. Reduction in percent change in the F scenario is observed with a decrease in the surface
runoff, baseflow and soil moisture content within a range of (1.5 to 12) % , (4.9 to 32) % and (-2.5 to 21) respectively followed by increase in ET within a range of -(1.8 to 3.5) %. The increase in the percent change in the GL scenario for all the components are lesser than the




CL scenario and the decrease in the percent change is lesser than the F scenario. Soil moisture content for all the scenarios shown in the boxplot below are obtained by summing up the soil water content for three defined soil layers. But an inspection for each layer (Not shown in Figure 8) shows that the soil moisture content in the top thin layer decreases in the NF, FF and  CL scenarios and increases in the F scenario whereas the soil moisture content in the second and third layer increases in the NF, FF and CL scenarios and decreases in the all F scenario.

Table 4 shows annual water balance components averaged over 20 years (1990-2010) for all scenarios. M6, M20 and M67 in the table indicates 3 models lying within 25th to 75th percentile among the ensemble of calibrated models. Models lying within 25th percentile and 75th percentile varies from one catchment to another i.e. NSE value varies. Therefore 25th and 75th percentile of NSE values have been computed for each catchment in the calibration period and 3 models have been chosen where NSE of all the subcatchments lie within 25th and 75th percentile to analyse the annual water balance averaged over 20 years. VIC model solves the water balance within the catchment using the Eq. (1) (Gao et al., 2010).

$$\Delta s/\Delta t = P - E - R - B , \tag{1}$$

where P, E, R and $\Delta s/\Delta t$ are the precipitation, evapotranspiration, runoff, baseflow and change of water storage respectively. Table 4 indicates that the model is able to estimate all the water budget components and maintain proper closure of the water balance in all the scenarios. Note that the average annual precipitation is 1318 mm, which is kept constant for all the scenarios.

### 4.4 Model parameter uncertainties

An ensemble of model parameter sets is derived for the entire basin based on pareto ranking to account for the uncertainties in the hydrological components owing to the land use change. The differences between the minimum and maximum percent changes in annual extreme flows and other water balance components in all the scenarios represents uncertainty in the simulated land cover change impacts by the selection of model parameters. Next, we seek to understand the interactions (Figure 10) among the model parameters in relation to the obtained annual water budget components (Runoff, ET, Baseflow and Soil moisture content) averaged over 20 years for the far future scenario at Kantamal. We choose Kantamal as it is

the best performing subcatchment among all with the highest NSE. Figure 10 shows the
behaviour of these model parameters within their respective ranges which results in

predictive uncertainties through the change in hydrological processes occuring within the
basin. Higher values of second layer of soil, (d2), i.e thicker depths, retaining more soil water,
produced more ET, hence resulting in less runoff and vice versa. There is also a trend of
thinner d2 generating more baseflow, however not as clearly identifiable as in case of runoff.
Higher values of parameter, binf, decreases the infiltration capacity of the soil and increases

the runoff, thereby reducing the subsurface drainage, soil moisture content and water
available for ET. Exponent of the unsaturated hydraulic conductivity parameter, Exp controls
the vertical drainage between the soil layers. Higher values of Exp decreases the drainage
between the soil layers for the same soil moisture content consequently decreasing the
baseflow generation and eventually resulting in more evaporation. Parameters, Ds and Ws

seem to have opposite influence over obtaining high soil moisture content, ET and baseflow
i.e either higher values of Ds interacting with lower values of Ws or lower values of Ds
interacting with higher values of Ws to increase the soil moisture content and producing more
ET and baseflow. However, baseflow related parameters, Ds, Ws and Dsmax are not clearly
identifiable in case of producing runoff. There is a clear pattern of lower values of velocity

producing more runoff and baseflow, and vice versa.

## 5. Discussion

We carried out sensitivity analysis with 3 objective functions to account for all parameters
sensitive to all the flow processes occuring within the basin. However, the objective function
used for calibration is based on the application of the model as also followed by Muleta and

Nicklow, (2005). Muleta and Nicklow,(2005) also implemented a calibration strategy guided
by global sensitivity analysis which reduced the uncertainties in streamflow. In overall,
parameter sensitivity results are in accordance with the findings of Demaria et al., (2007)
where the parameters Exp and binf were sensitive to the objective function RMSE (focusses
on high flows) and parameters, d1 and ksat were slightly sensitive to the streamflow.

Owing to very high computational costs, we abstained from modelling the land cover change
impacts at each subcatchments individually. Given that the subcatchments are of conflicting
catchment characteristics, there is no feasible point that optimizes all five parameter sets
obtained from different subcatchments. Therefore, the idea of pareto ranking which has been

applied in many studies (Gupta et al., 2003; Shi et al., 2008) serves the purpose of generating

a common set of models by looking for an acceptable trade-off among the calibrated models of all the subcatchments. Our calibration results suggest that the models tend to perform better at the bigger catchments and yielded lower NSE values at the smaller catchments, Salebhata and Sundergarh. These findings are in agreement with some literatures (Kneis et al., 2014; Mishra et al., 2008; Nayak, Venkatesh, Thomas, & Rao, 2010) wherein the calibrated

hydrological models yielded low NSE values at Salebhata or Sundergarh. This is possibly because errors in the input data, observed data or the errors resulting from uncertain parameters counter-balanced each other at the bigger catchments unlike the smaller catchments.

LUH2 is a new dataset, not yet extensively used in basin-scale hydrology. The major changes

occuring within the basin in the future scenarios as predicted by LUH2 are expansion of cropland at the expense of forest, particularly DBF which also agrees with a recent study carried out in the Mahanadi river basin by Behera et al., (2018). A recent study by Krause et al., (2019) predicted  worldwide increment in runoff (67%) and a variable responses of ET across different scenarios under the effects of land use and climate changes using LUH2

dataset.

Due to some unknown reasons, slight differences in extreme flows are observed between Near Future and Far Future despite a substantial increment in cropland occuring at all the subcatchments in the Far Future relative to the Near Future.  Despite a significant change in land cover occuring within the basin in the future, it should be noted that these changes are

not consistent at all the subbasins (See Table S3 in Supplementary section). Basantpur is the biggest subcatchment which is projected to undergo a maximum expansion in cropland areas (37%) among all the subcatchments. Therefore, future scenarios should reflect more percent change in extreme flows at Basantpur. This is to some extent, supported by our result (Figure 8) as the maximum percent  increase in extreme flows from within the range of best ensemble

models  are  observed  at  Basantpur (21%) followed by Kantamal (13%), Kesinga (12%), Salebhata (10%), Sundergarh (14%). However, the median values at all the subcatchments suggest a percent increase in extreme flows of almost same order (2- 5) %. This can be attributed to the fact that the  effects of LULC change on streamflows in a large catchment are masked out by other factors, such as the spatial variability in precipitation, heterogeneous

soils etc. (O'Connell et al., 2007). This result agrees with the findings of Wilk and Hughes,



(2002) wherein removal of large forests led to little or no changes in annual runoff in large heterogeneous catchments in South India. LULC effects on floods although vary with the catchments scales, however, identifying any hydrological changes becomes difficult as the catchment size increases (Hurkmans et al., 2009; Rogger et al., 2016; Viglione et al., 2016).

Removal of forests at the expense of cropland decreases the LAI of the natural vegetation and hence decreases ET. Moreover, removal of forest cover reduces the root water uptake by plants which increases the water content of the second and third layer of the soil. The top thin soil layer in VIC model helps in partitioning the rainfall amount into direct runoff and the amount entering the soil. Therefore, the increase in the cropland results in more direct runoff

thus reducing the soil moisture content in the first soil layer. Since surface runoff in VIC model is caused by saturation, increases in the surface runoff is correlated with the increase in soil moisture (Hurkmans et al., 2009). This agrees well with our result (Figure 9) where the increase in the runoff is in correlation with total soil moisture content. The expansion in the cropland results in more impervious area thereby augmenting the surface runoff. The

increase in surface runoff and decease in ET resulting from the agricultural expansion at the expense of forest, are in line with the findings of other studies in the Mahanadi river basin in India, neighbouring basins and elsewhere (Abe et al., 2018; Berihun et al., 2019; Cornelissen et al., 2013; Costa et al., 2003; Dadhwal et al., 2010; Das et al., 2018; Kundu et al., 2017). Kundu et al., (2017) found an increase in runoff and decrease in ET due to the expansion in

projected agricultural land in Narmada river basin in India. Das et al., (2018) predicted that deforestation, urbanization and cropland expansion in eastern river basins of India, in the future would increase runoff and baseflow and decrease ET.

The impacts on the annual water balance of the entire basin is, however, small in terms of magnitude. Research elsewhere (Ashagrie et al., 2006; Fohrer et al., 2001; Kumar et al., 2018;

Patidar and Behera, 2019; Rogger et al., 2016; Wagner et al., 2013; Wilk and Hughes, 2002) have also reported that the impacts of land cover change on water balance components in a large scale river basin are too small to be detected due to the compensation effects. Patidar and Behera, (2019) in a recent study in a large river basin in India, showed that the impacts of land cover change on ET and runoff cancels out at the basin scale and reported that the

conversion of forest to agriculture may not alter the water balance significantly. Our result shows that a major portion of precipitation is contributing to ET in all scenarios (Table 4) which is consistent with Das et al., (2018), Garg et al., (2019) and the impact of land cover change

on ET is more than other water balance components which agrees with the findings of Kundu et al., (2017) in a river basin in India. Moreover, (Garg et al., 2019) found that croplands

contributes more to ET than streamflow in a river basin of a similar climate zone in India. The changes in extreme flows and the water balance components are more pronounced in the hypothetical scenarios. All cropland (CL) scenario showed the maximum increase in flows whereas All Forest (F) scenario resulted in the maximum reduction in flows which is in line with previous studies (Ma et al., 2010; Mishra et al., 2010; Wilk and Hughes, 2002). Wilk and

Hughes, (2002) reported that the largest increase in runoff resulted from total conversion of the basin to agriculture in in South India. Maximum reduction and increment in baseflow are observed in F and CL scenario respectively which is consistent with the observations in other studies in terms of direction of change (Mishra et al., 2010, Vano et al., 2006). However, unlike our study, most of these studies reported that an increase in runoff in Grassland (GL) scenario

is more than the cropland scenario. These differences can be attributed to the process of generating the hypothetical scenarios. For instance, the forest scenario in Ma et al., (2010) is represented by converting all grassland, barren lands and Croplands only above a certain elevation whereas Mishra et al., (2010) framed the hypothetical scenarios by converting a single grid cell to 100% Cropland, Forest and Grassland. However, more emphasis should be

given to the CL scenario as this relates to the major changes occurring in the basin as per the future projected LUH scenarios.

Multiple parameter sets can yield equally good or acceptable model outputs due to the complex interactions among the parameters , known as equifinality, considered as one of the main sources of uncertainty in hydrological modelling (Her et al., 2019). Taking this into

consideration, the methodological approach in our study involves Monte Carlo based model calibration guided by parameter sensitivity analysis to generate an ensemble of best performing models rather than one behavioural model. Despite, the median percent change in the annual extreme flows indicate no significant change at the subcatchments (on the order of +3%), we show that the maximum increase in extreme flows within the uncertainty bound

in the FF scenario at Basantpur is 20.5%. This indicates that even a small set of calibrated models can predict a wide range of flows through different hydrological processes occurring within the basin. For instance, the reverse interactions among Ds and Ws, lead to the varying hydrological processes occuring within the basin, thereby affecting the partition of water in the soil column. Similar results are found in Eum et al., (2016) wherein the VIC model


simulations using different calibrated parameter sets had led to higher uncertainties in the annual peak flows. Her et al., (2019) demonstrated that the model parameter uncertainties are significant for some hydrological components. Furthermore, the range of these hydrological estimates provides more straightforward and explicit quantification of uncertainty than other statistical measures such as variance, interquartile ranges (Her et al.,

2019). Perhaps, the most important implication of this study is that the changes that are likely to occur can have negligible to significant impacts on hydrology of Mahanadi river basin.

## 6. Conclusion

India is a fast developing and second most populated country in the world. To cater a huge population, there has been a substantial change in the land cover types (e.g. Agricultural

intensification, industrial expansion etc.) in the last decades, playing a significant role in altering the basin's hydrology. The purpose of this study is to quantify the hydrologic response of the subcatchments of Mahanadi river basin to the land cover changes in the present and the future through the implementation of a well calibrated physically distributed hydrological model. Considering the changing environment across the globe, the usage of a physically

distributed hydrological model is more appropriate than the traditional empirical approach (Garg et al., 2019). The VIC model has been set up and run for the Mahanadi river basin on a regional scale of 5 kms and the modelled discharge are well in agreement with the measured discharge at all the subcatchments considered. The methodological approach used in this study helps to comprehend the possible impacts of changing land cover scenarios within a

modelling framework of detailed calibration and sensitivity analysis. The calibration of sensitive model parameters has resulted in more realistic ensemble model simulations as it accounts for the model parameter uncertainties in quantifying the impacts of land cover changes. Deforestation at the expense of cropland dominated the land cover change processes in the study area which have implications on the hydrological processes.

Some major findings from this study are:

1. Most influential model parameters across all the subcatchments and objective functions are unsaturated hydraulic conductivity parameter, Exp and baseflow parameter, Dsmax. Non-influential parameters that are exempted from calibration


are the first depth of soil layer, d1, saturated hydraulic conductivity, ksat and routing parameter, diff

2. Future LULC scenario, RCP3.4 SSP4 from LUH indicates cropland and forest remains the major land cover types in the basin with a noticeable increase in the cropland (23.3%) at the expense of forest (22.65%) by the end of year 2100 compared to the

baseline year, 2005.

3. In overall, results from the best ensemble models indicate, conversion of forested areas (32072 km$^2$) to agriculture (32759 km$^2$) has led to reduced LAI which in turn reduces the ET and increase runoff and baseflow in the future.

4. LULC changes that are likely to occur in the future can have negligible (2%) to

significant impacts (20%) on the extreme flows of Mahanadi river basin.

5. The hypothetical conversion of all the forest and grassland areas to the cropland results in maximum increase in the annual extreme flows.

The ensemble models obtained, thus, account for the parameter uncertainties while

predicting a wide range of plausible hydrological changes resulting in a more robust and reliable analysis, which shall make the land cover change mitigation strategies and water resources management plans more effective. From this analysis, it can be understood that the recurrent flood events occurring in the Mahanadi river basin might be influenced by the changes in LULC at the catchment scale. This study also provides valuable insights about the

sensitivity of the model parameters to the model output and the interactions among the model parameters in producing the changes in hydrological behaviour for same land cover change. The present study focusses on the altered hydrological responses only owing to the changes observed in the land cover assuming the climatic variables constant for all the LULC scenarios. Climate also changes with time and hence this assumption does not consider the

actual condition on the ground. It is noteworthy to mention that previous studies in this basin led by Asokan and Dutta, (2008) and Ghosh et al., (2010) had predicted an increase in the peak flows in the monsoon months at the 2100's which if combined with the impacts resulting from land cover changes might result in adverse flooding in the basin. Therefore, future studies shall focus on modelling the combined impacts of climate and land cover changes on

hydrology of Mahanadi river basin, considering model parameter uncertainty, which is currently lacking in many studies.



*Data availability.*
Dem is freely available from https://bhuvan-app3.nrsc.gov.in/data/download/index.php. Unit
Hydrograph is adopted from https://vic.readthedocs.io/en/vic.4.2.d/Documentation/Routing/UH/.

Daily gridded rainfall, maximum and minimum temperature are freely available from
http://www.imdpune.gov.in/Clim_Pred_LRF_New/Grided_Data_Download.html.  Wind speed data
is freely available from
https://psl.noaa.gov/cgibin/db_search/DBSearch.pl?Dataset=NCEP+Reanalysis+Daily+Averages
LUH2 datasets are downloaded from https://luh.umd.edu/data.shtml. Observed discharge data are

available from http://cwc.gov.in/. The source code for VIC-3L version 4.2.d is available from
https://github.com/UW-Hydro/VIC/releases/tag/VIC.4.2.d.

*Competing interests.* The authors declare that they have no conflict of interest.

*Author's Contribution.* SN, MRR and RR designed this study. SN performed the model
simulations. MR and RR assisted SN in analysing and discussing the results. SN wrote the
manuscript and MR and RR commented on the manuscript.

*Acknowledgements.* This work was partially funded by the BEMUSED project funded by
Natural Environment Research Council (NERC; grant number NE/R004897/1)  and by the
International Atomic Energy Agency of the United Nations (IAEA/UN) under the Coordinated
Research Project (CRP D12014).

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





**Table 1.** VIC-3L model parameters considered for the sensitivity analysis

| | Parameter | Realistic range | Estimating method |
|---|---|---|---|
| Soil parameters | Initial soil moisture (mm) | 56.7-276.4 | (Cosby et al., 1984; Reynold et al., 2000; Rawls et al., 1998 ) |
| | Bulk density (kgm$^{-3}$) | 1390-1520 | |
| | Fractional soil moisture content at critical point : Wcr_frac | 0.02-0.46 | |
| | Fractional soil moisture content at wilting point: Wp_frac | 0.01-0.23 | |
| | Thickness of first soil layer: d1 | 0.01-0.3 | Subjected to SA / Calibration |
| | Thickness of second soil layer: d2 | 0.3-3.5 | |
| | Thickness of third soil layer: d3 | 0.3-3.5 | |
| | Max. velocity of baseflow: Dsmax | 0.0001-30 | |
| | Fraction of max. velocity of baseflow: Ds | 0.0001-1 | |
| | Parameter to describe the Variable Infiltration Curve: binf | 0.0001-4 | |
| | Saturated hydraulic conductivity: Ksat | 1-5000 | |
| | Fraction of maximum soil moisture of the third layer: Ws | 0.0001-1 | |
| | Parameter characterizing the variation of saturated hydraulic conductivity with soil moisture: Exp | 3.1-30 | |
| Vegetation Parameters | Root zone depth (m) | 0-2 | Raje et al. 2014, Zeng 2001 and Nijssen et al. 1997 |
| | Root fraction for each land cover type | 0-1 | |
| | Albedo: α | 0-0.168 | GLDAS database http://ldas.gsfc.nasa.gov/ |
| | Architectural Resistance: $r_{arc}$ (sm$^{-1}$) | 0-60 | |
| | Roughness length: $z_0$ (m) | 0-2.65 | |
| | Displacement height: $d_0$ (m) | 0-27.37 | |
| | Minimum stomatal resistance: $r_{min}$ (sm$^{-1}$) | 0-150 | |
| | Leaf Area index, LAI | 0-3.2 / 0-10.3 | GLDAS/ MODIS |
| Routing parameters | Velocity | 0.1 - 3 | Subjected to SA / Calibration |
| | Diffusion | 500 -5000 | |

**Table 2**. Percent of each Land use type in VIC2005 and LUH2005 (WB – Water Body; ENF – Evergreen Needleleaf Forest; DBF – deciduous Broadleaf Forest; GL- Grassland; CL- Cropland; U – Urban)

| LULC classes (%) | NRSC2005 | LUH2005 |
|---|---|---|
| WB | 2.6 | 0.76 |
| ENF | 0.08 | 0 |
| DBF | 35.98 | 41 |
| GL | 0.13 | 4.7 |
| CL | 49 | 53 |
| U | 0.52 | 0.4 |
| BG | 12.3 | 0 |







**Table 3:** Land cover area change in Mahanadi river basin

| LULC classes (%) | Baseline 2005 | Present 2015 | Near Future 2050 | Far Future 2100 | All Cropland | All Forest | All Grassland |
|---|---|---|---|---|---|---|---|
| WB | 0.76 | 0.76 | 0.76 | 0.76 | 0.76 | 0.76 | 0.76 |
| F | 41 | 42.4 | 25.45 | 18.35 | 0 | 97.7 | 0 |
| GL | 4.7 | 4.6 | 6.9 | 4.5 | 0 | 0 | 97.7 |
| CL | 52 | 51.5 | 65.6 | 75.3 | 97.7 | 0 | 0 |
| U | 0.4 | 0.5 | 1.2 | 1 | 0.4 | 0.4 | 0.4 |


**Table 4.** Annual water balance components (runoff, ET, baseflow, change of water storage) for all scenarios (LUH: baseline, present, near future, far future; hypothetical: cropland, forest and grassland) computed for 3 models (M6:Model 6, M20:Model 20, M67:Model 67) out of 101 models lying within 25[th] percentile and 75[th] percentile of NSE values computed for each

catchment. Annual water balance components are averaged over 20 years (1990-2010)

| Scenarios | LULC area (%) | | | Runoff/year (mm) | | | ET/year (mm) | | | Baseflow/year (mm) | | | Change of water storage/year (mm) | | |
|---|---|---|---|---|---|---|---|---|---|---|---|---|---|---|---|
| | CL | F | GL | M6 | M20 | M67 | M6 | M20 | M67 | M6 | M20 | M67 | M6 | M20 | M67 |
| Baseline | 53 | 41 | 5 | 265 | 333 | 290 | 981 | 948 | 976 | 66 | 32 | 30 | 4 | 4 | 21 |
| Present | 52 | 42 | 4 | 266 | 334 | 290 | 980 | 947 | 975 | 67 | 32 | 21 | 4 | 4 | 31 |
| Near Future | 66 | 25 | 6 | 277 | 345 | 302 | 961 | 932 | 957 | 74 | 36 | 36 | 4 | 4 | 22 |
| Far Future | 75 | 18 | 4 | 280 | 348 | 305 | 957 | 929 | 953 | 75 | 36 | 37 | 4 | 4 | 22 |
| Cropland | 97 | 0 | 0 | 288 | 354 | 314 | 949 | 924 | 944 | 77 | 35 | 36 | 4 | 4 | 23 |
| Forest | 97 | 0 | 0 | 238 | 308 | 260 | 1021 | 978 | 1014 | 54 | 28 | 24 | 5 | 4 | 19 |
| Grassland | 97 | 0 | 0 | 271 | 336 | 296 | 974 | 946 | 969 | 68 | 32 | 31 | 4 | 3 | 21 |




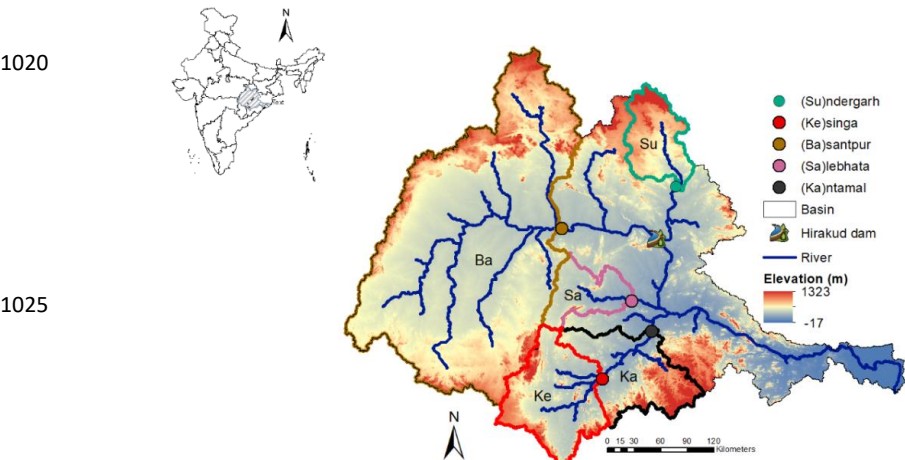

**Figure 1:** The Mahanadi river basin boundary and its subcatchments (indicated with Ba, Ka, Ke, Su, Sa).




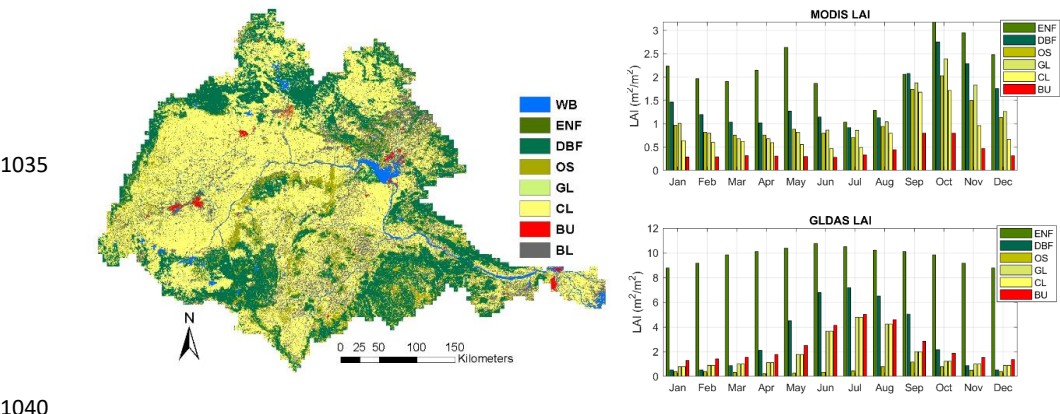

**Figure 2: (a)** LULC map of Mahanadi river basin **(b)** LAI from MODIS and GLDAS





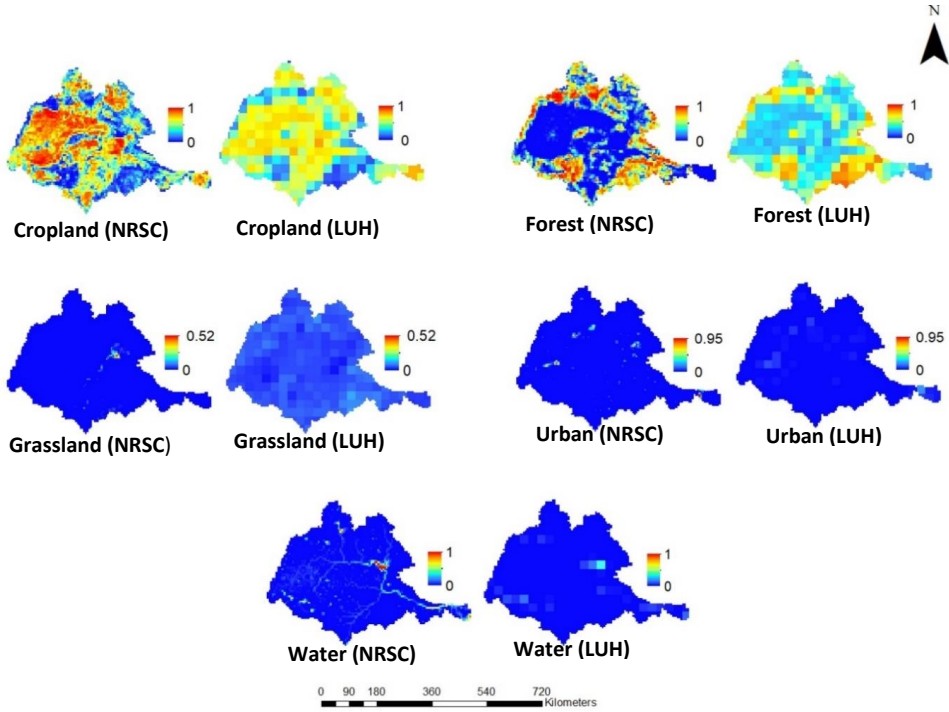


**Figure 3.** Comparison of LULC maps from NRSC and LUH for 2005



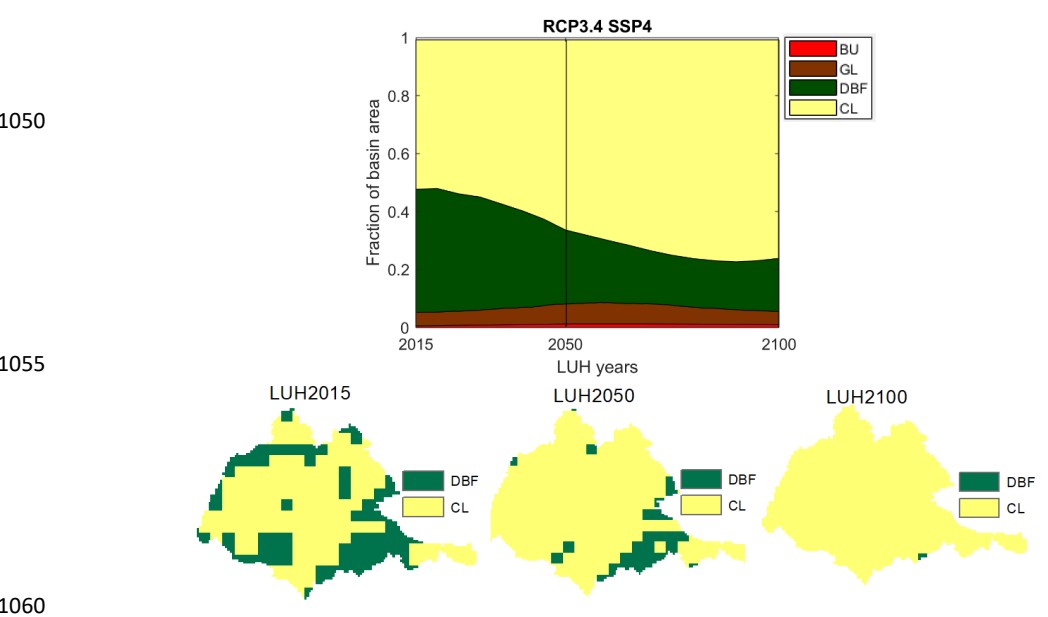


**Figure 4:** Top: Fraction of catchment area occupied by Land use classes for scenario RCP3.4 SSP4 Bottom: land cover scenarios from LUH (resolution- 25 km) for years 2015, 2050 and 2100 used in this study. LUH land cover classes shown here are resampled to the model grid resolution and only the predominant class is shown here for clarity. For actual model simulations VIC accounts for the individual proportion for each land cover type at each grid point.



**Figure 5:** Sensitivity indices (normalised mean) of EET method for VIC model parameters for all the subcatchments and objective functions. Colour bar on the right side indicates sensitivity of the model parameters to the streamflow. '0' indicates least sensitive and '1' indicates most sensitive.



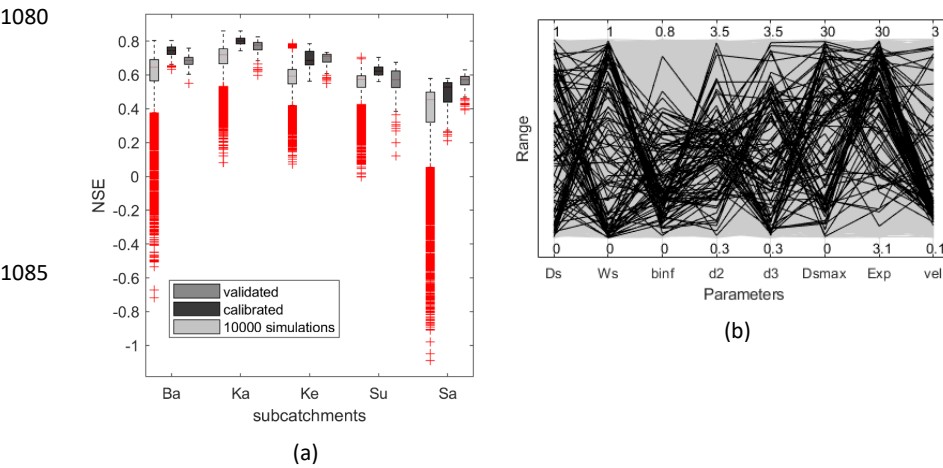


**Figure 6: (a)** Box plot showing NSE range for 10,000 simulations, calibrated (best) simulations obtained through pareto ranking and validated simulations **(b)** Parallel coordinate plot showing soil parameters that had resulted in best simulations during model calibration.





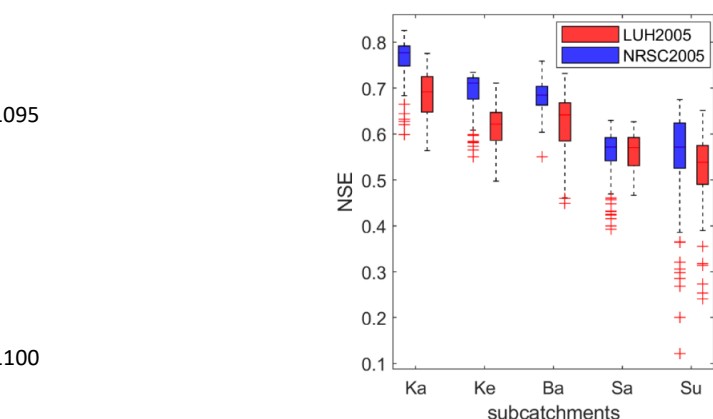

**Figure 7:** Box plot showing comparison of performance of model simulations: control scenario
using LUH2005 and calibrated best simulations using NRSC2005 for the validation period
1105    (2001-2010)



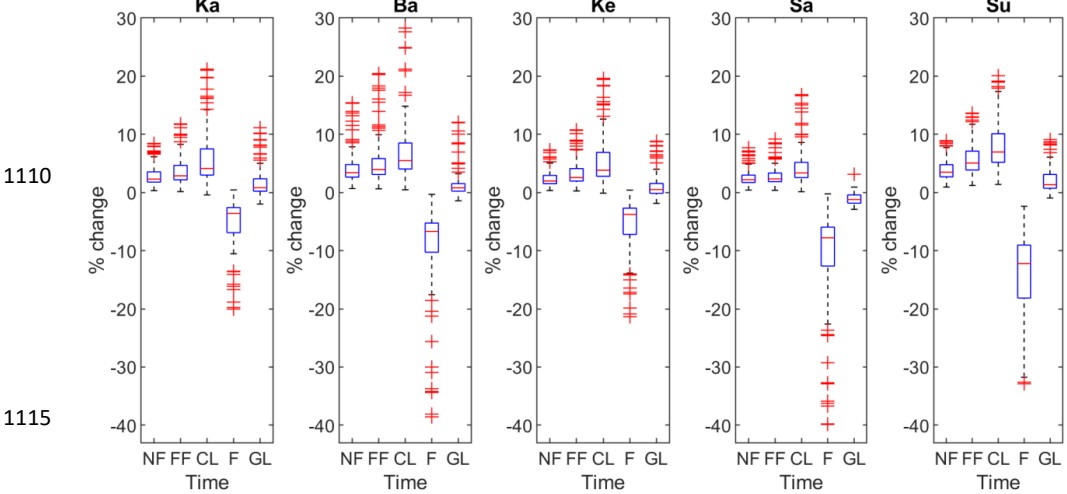

**Figure 8:** Percent change in annual average of extreme flows (i.e., 95th percentile or higher)
in the Near future (NF), Far future (FF), Cropland (CL), Forest (F) and Grassland (GL) scenarios
with respect to baseline land cover condition from 2005 for all the subcatchments. Please
note that the climate forcing is kept fixed for the period corresponding to year (1990-2010)
The results are shown for the 101 'best' model simulations obtained through calibration.



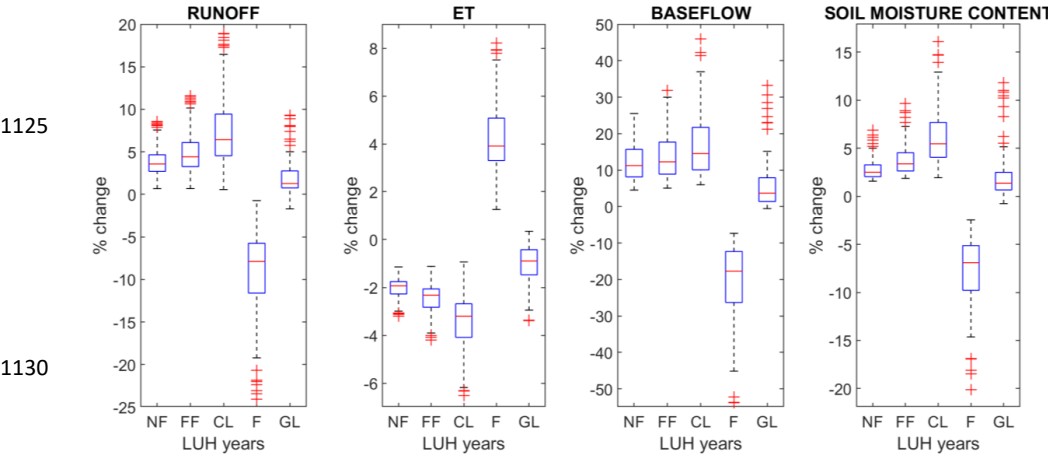

**Figure 9:** Box plot showing percent change in mean annual Runoff, ET and baseflow and change in Soil moisture for (1990-2010) for the scenario RCP3.4 SSP4 for the entire Mahanadi river basin. Please note that the climate forcing is kept fixed for the period corresponding to year (1990-2010) and therefore precipitation is constant in all the scenarios considered.

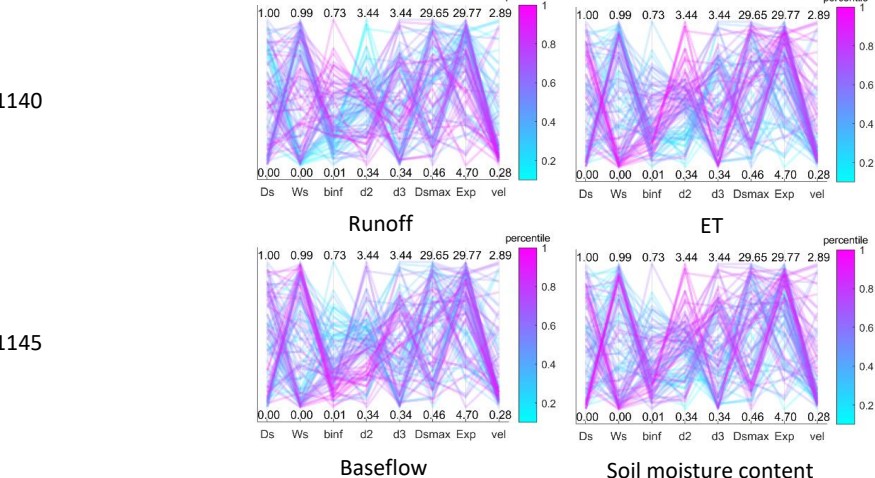

**Figure 10:** Parallel coordinate plot showing model parameters that had resulted in lowest to highest estimates of water balance components in the Far Future scenario for subcatchment Kantamal. Each line represents corresponds to a simulation performance: '0' in the colour bar



represents the lowest performance value and '1' represents highest value obtained from best

101 simulations.