# Peer review of "Quantifying the impact of land cover changes on hydrological extremes in India"

_Hydrology and Earth System Sciences, 2020_

## Referee Comment (RC1) · Anonymous Referee #1 · 2 Aug 2020

**1   Overview**

The study by Naha et al. aims to assess the role of future land use land cover (LULC) change in altering the nature of hydrological extremes in the Mahanadi river basin in India, under different future scenarios, described by a combination of socio-economic pathways (SSPs) and representative concentration pathways (RCPs). The manuscript deals with an important topic in a region sensitive to hydrological extremes. It is, in general, well-written and easy-to-read. However, several aspects of the study, including novelty, introduction, and methodology (specifically the sensitivity analysis and calibration) are either not well designed or incompletely described. I cannot recommend the publication of this study until these issues are satisfactorily addressed.

[Figure]

**2   Major Comments**

1. **Introduction**: One of the main issues is the lack of novelty in the presented research, outside its contribution as a case study. The authors need to clearly bring out the novelty of their research compared to the existing research (cited in the study) which have already shown that model parameters have an impact on future land use change studies.

   - **Line 50 - 55**: "However, the exact role of LULC changes in modifying river discharge is still elusive (Rogger et al., 2016) and therefore, remains a challenge to isolate the sole impacts of land use changes on hydrology of a river basin (Tsarouchi and Buytaert, 2018)". These claims are very vague. What are the specific challenges and how does the present study aims to overcome them?
   - **Line 105 - 110**: What is missing in these studies which the authors have solved in the presented study? This is not very clear.
   - **Line 115 - 120**: The research questions are very vague. None of the research questions pertain to the central question question: impact of LULC change on hydrological extremes. Instead it focuses on the modeling part.

2. **Design of Study and Methodology**: I have serious issues with the design of the study. As it stands, it reads like two different studies which are unrelated to each other: 1) quantify impact of parameter uncertainty and 2) quantify impact of lulc cover change. However, there is no attempt to connect these two objectives. It is well known that parameter uncertainty will have an impact on the output of hydrologic models. However, how does this specifically relate to LULC change studies? This is not clear

   - It is not clear as to why only soil parameters are selected for sensitivity analysis. I would imagine that the selection of the hydrologic model should be

based on how sensitive the model parameters are to vegetation parameters which LULC. In fact, this should be the first step to understand the behavior of the model to changes in LULC. This information is missing.

- Model calibration and Pareto ranking is poorly described. What are the specific steps involved in assigning the ranks?

- I am not sure if I understood this correctly but only a range of parameter values are provided for each LULC type. It would be better to provide the parameter values that are used for each LULC type to see if they are physically consistent.

3. **Results, Discussion and Conclusions**:

- **Figure 6b**: The calibrated parameter ranges are too wide (equifinality).I am not sure why calibration was not able to narrow the range of parameters here. The authors need to discuss why the ranges are not more constrained even after calibration. The discussion in lines 615 - 635.

- **Figure 9**: The issue of using only related parameters for calibration is reflected in the water balance components. Evaporation, which I assume is dependent on vegetation parameters, shows very little change (less than 8 %), while soil moisture exhibits changes large variations. This makes it very difficult to discern the impact of LULC changes alone. I recommend that the authors also present results with only the best parameter. Of course, I assume here that actual evaporation is explicitly modeled in VIC and is not an input.

- **Line 655 - 670** - Many of the points under major findings are a repetition of the results and do not represent substantial or novel conclusions.

---

## Author Comment (AC1) · 26 Jan 2021

Reviewer 1

We appreciate the comments and insights provided by Reviewer#1, and below in **bold** include our response to the comments.

**General Overview:**

The study by Naha et al. aims to assess the role of future land use land cover (LULC) change in altering the nature of hydrological extremes in the Mahanadi river basin in India, under different future scenarios, described by a combination of socio-economic pathways (SSPs) and representative concentration pathways (RCPs). The manuscript deals with an important topic in a region sensitive to hydrological extremes. It is, in general, well-written and easy-to-read.

**We would like to thank the reviewer for this positive comment.**

However, several aspects of the study, including novelty, introduction, and methodology (specifically the sensitivity analysis and calibration) are either not well designed or incompletely described. I cannot recommend the publication of this study until these issues are satisfactorily addressed.

**We have now completely redesigned this study based on the reviewer's suggestions and will be rewriting a significant portion of the manuscript, highlighting its novelty, and providing more clarity to the sections mentioned above.**

**Major Comments:**

1. Introduction:

One of the main issues is the lack of novelty in the presented research, outside its contribution as a case study. The authors need to clearly bring out the novelty of their research compared to the existing research (cited in the study) which have already shown that model parameters have an impact on future land use change studies.

• Line 50 - 55: "However, the exact role of LULC changes in modifying river discharge is still elusive (Rogger et al., 2016) and therefore, remains a challenge to isolate the sole impacts of land use changes on hydrology of a river basin (Tsarouchi and Buytaert, 2018)". These claims are very vague. What are the specific challenges and how does the present study aim to overcome them?

• Line 105 - 110: What is missing in these studies which the authors have solved in the presented study? This is not very clear.

• Line 115 - 120: The research questions are very vague. None of the research questions pertain to the central question: impact of LULC change on hydrological extremes. Instead, it focuses on the modelling part.

**We thank the reviewer for suggesting more clarity in highlighting the novelty. As we have redesigned the study, we will modify the Introduction section of our manuscript highlighting the novelty and reframing the research questions with lot more clarity.**

**We think the reviewer is trying to point out the citations, (Chaney et al., 2015; Singh et al., 2014; Zhang et al., 2019) by saying 'existing research (cited in the study)'. These are some studies that highlighted the need of considering model parameter uncertainties while modelling the changes in hydrology (not particularly in context of land cover change studies). Our intention of adding these citations was to highlight that although much attention has been paid to the model predictive uncertainties, less for uncertainties in the context of impact assessment of land use change. For instance, several studies (Bennett et al., 2018; Feng and Beighley, 2020; Her et al., 2019) exist that takes into account model parameter uncertainties while dealing with climate change, however rarely, it has been considered while assessing land cover changes.**

**We agree that the way in which the citations are used in these sentences in the original manuscript are unclear. We will reframe the sentences along with some added citations to prove our statement as we mentioned in the paragraph above.**

2. **Design of Study and Methodology:**

I have serious issues with the design of the study. As it stands, it reads like two different studies which are unrelated to each other: 1) quantify impact of parameter uncertainty and 2) quantify impact of lulc cover change. However, there is no attempt to connect these two objectives. It is well known that parameter uncertainty will have an impact on the output of hydrologic models. However, how does this specifically relate to LULC change studies? This is not clear.

**We thank the reviewer for pointing this out. As we explained above, we will re-frame the objectives of the paper and will add necessary explanations and figures (if required) to better associate the land cover change impacts with the model parameter uncertainties.**

• It is not clear as to why only soil parameters are selected for sensitivity analysis. I would imagine that the selection of the hydrologic model should be based on how sensitive the model parameters are to vegetation parameters which LULC. In fact, this should be the first step to understand the behaviour of the model to changes in LULC. This information is missing.

**We thank the reviewer for pointing this out. This was an important feedback provided by the reviewer. We realize this was the shortcoming in our study and now we have taken the vegetation parameters into account. We are re-running the Sensitivity Analysis (SA) and model calibration experiments. We have now conducted some global sensitivity analysis experiments including the vegetation parameters and a few more soil-related parameters (Figure 1). Our preliminary results suggests that canopy height is not sensitive, and as roughness length and displacement height are both computed from canopy height directly (following standard meteorological approaches), we have decided not to include these parameters in our final experiments. We derived daily Leaf Area Index (LAI) product directly from MODIS AQUA/TERRA (2000-2015), hence not included as a parameter. We also redesigned the root zone allocation in our study. In our new approach shown in Figure 2, the root zone fractions for each root zone depths and fractions varies more realistically with respect to the soil depths while calibrating. In total, 16 parameters including soil, vegetation and routing parameters are subjected to sensitivity analysis in the revised version and our**

**preliminary results suggest that re-designing our SA and calibration experiments have improved our model performance.**

• Model calibration and Pareto ranking are poorly described. What are the specific steps involved in assigning the ranks?

**We are now re-running the calibration experiments and the Pareto Ranking approach is no longer applied**.

• I am not sure if I understood this correctly but only a range of parameter values are provided for each LULC type. It would be better to provide the parameter values that are used for each LULC type to see if they are physically consistent.

**We thank reviewer for this suggestion, and we will incorporate this in our revised manuscript. We will add a table in the supplementary section showing the vegetation parameter values or ranges (if subjected to sensitivity analysis) along with their sources, used in this study for each dominant LULC types in the basin.**

[Figure]

**Figure 1: Sensitivity indices of EET method for VIC model parameters for all the sub-catchments. Colour bar on the right side indicates sensitivity of the model parameters to the streamflow. '0' indicates least sensitive and '0.6' indicates most sensitive. Mean indicates direct effect of these parameters on model output. Standard Deviation indicates parameters interaction effects on model output. Vegetation parameters marked in 'solid black' rectangle considered for SA and marked in 'dash black' are not considered.**

[Figure]

**Figure 2: (left) Representation of rooting distributions (typically kept fixed) in VIC-3L model. $z_1$, $z_2$ and $z_3$ are the user-defined depths of three root zones, respectively. $d_1$, $d_2$, $d_3$ are the depths of three soil layers. $f_1$, $f_2$ and $f_3$ are user-defined fractions of root in each zone, respectively. $f_1'$, $f_2'$ and $f_3'$ are fractions of root in each soil layer computed by VIC. We used this approach in the original manuscript. (right) Our new approach of representation of rooting distributions in VIC-3L model proposed in the revised version. $z_r$ is the total root depth which is fixed. However, $z_1$, $z_2$, $z_3$ varies with $d_1$, $d_2$ and $d_3$ respectively. $f_1$, $f_2$, $f_3$ are the root fractions in respective root zones computed using** (Zeng, 2002)**.**

**3. Results, Discussion and Conclusions:**

• Figure 6b: The calibrated parameter ranges are too wide (equifinality). I am not sure why calibration was not able to narrow the range of parameters here. The authors need to discuss why the ranges are not more constrained even after calibration. The discussion in lines 615 - 635.

• Figure 9: The issue of using only related parameters for calibration is reflected in the water balance components. Evaporation, which I assume is dependent on vegetation parameters, shows very little change (less than 8 %), while soil moisture changes exhibit large variations. This makes it very difficult to discern the impact of LULC changes alone. I recommend that the authors also present results with only the best parameter. Of course, I assume here that actual evaporation is explicitly modelled in VIC and is not an input.

**We have made substantial changes in the model sensitivity and calibration methods which we are reanalysing now, and our preliminary results indicate improvement in the model performance (not shown). Hence, we believe these changes will have impact on the issues raised by the reviewer in these particular figures. We will add necessary explanation and discussion in this regard in the revised manuscript.**

• Line 655 - 670 - Many of the points under major findings are a repetition of the results and do not represent substantial or novel conclusions.

**We thank the reviewer for pointing that out and will take that into account in the revised version of the manuscript.**

---

## Referee Comment (RC2) · Anonymous Referee #2 · 4 Feb 2021

General Comments:

The study investigates how the hydrological regime of the Mahanadi river basin would respond to the current and future land cover scenarios under a large-scale hydrological modelling framework. The recently released dataset LUH2, which has not yet extensively used in this study to provide current and future projected land cover scenarios. Although many studies have addressed the effects of future land cover changes on hydrological processes, I believe the novelty of the study lies with the consideration of parameter uncertainties of the physically-based model VIC, which are widely used by the hydrology community when evaluating LULC and climate changes. Valuable insights are provided into the sensitivity of the model parameters to the model outputs and the interactions among the model parameters in

producing changes of the hydrological regime within different LULC scenarios.

To some point, I agree with RC1 that the parameter sensitivity tests are not properly designed and there lacks through analysis to link the sensitivity results with the LULC impacts. However, the authors agreed to re-conduct the sensitivity analysis and the calibration experiments with the involvement of more vegetation and soil-related parameters. I am looking forward to the new outcomes and modifications.

Specific comments:

1. Line 68, "results in an increase in surface runoff and decreases river discharge", this trends seem to be contradictory between the surface runoff and the river discharge. Please check and confirm it.

2. Figure 2, please indicate in the figure title the time period of the LULC map and LAI data, and show the land use types in the legends rather than using abbreviations.

3. Line 198, explain what the exact meaning and the geographic scope of the "3 root zones".

4. Line 251, should "Klein-Gupta Efficiency" be "Kling-Gupta Efficiency (KGE)"? It is better if the equations of NSE, lnNSE and KGE could be given.

5. Line 256-257, "Parameters which showed poor performance when tested across all the subcatchments and objective functions were discarded." The sentence is unclear, please rewrite it to avoid misunderstanding.

6. Line 263-264, "the model was run, calibrated, and validated daily for each parameter set for the time period (1990-2000)". I think the authors only mean "calibrated" here, since the validation period is found to be 2001-2010 in the following manuscript.

7. Line 330-331, "In all the six cases of model run, meteorological forcing is held constant...", are the forcing data from the baseline scenario (2005) are used? Please give detail explanations.

8. Line 379-380, "The grey lines in the parallel coordinate plot in Figure 6b shows. . .", the grey lines are nearly invisible in Fig. 6b.

9. Line 384, change the title of "4.2 Control case scenario performance" to "Baseline scenario performance" to keep consistency with the rest expressions in the manuscript.

10. Line 421, the time period 1990-2010 is used for analyzing the effects of different LULC scenarios on the streamflow in the study. However, this covers the calibration and validation periods used for generating the VIC model parameters. In normal case, data used for calibrating models should not be used again for further analysis.

11. Line 469-471, "Reduction in percent change in the F scenario is observed with a decrease in the surface runoff, baseflow and soil moisture content within a range of (1.5 to 12) % , (4.9 to 32) % and (-2.5 to 21) respectively followed by increase in ET within a range of -(1.8 to 3.5) %." Values in the first three brackets should be negative (representing decreases) while values in the last bracket should be negative (representing increases).

12. Line 503-520: this paragraph describes how the parameters influences the change of the streamflow as well as other water balance components. However, these descriptions seem to be deductive from the physical definitions of the parameters, rather than being concluded from Fig 8. The results showed in Fig 8 are not well analysed and the sensitivity of parameters should be given in a more quantitative way.

13. Line 531-532, "Give that the subcatchments are of conflicting catchment characterisitic, there is no feasible point that optimizes all five parameter sets. . .", please explain what are the exact meaning of the "conflicting characteristics". I believe the rainfall-runoff characteristics of one subcatchment should not be so distinct from the others.

14. Line 635-636: "Perhaps, the most important implication of this study is that the changes that are likely to occur can have negligible to significant impacts on hydrol-

ogy of Mahanadi river basin". The expression of the sentence is unclear, especially "negligible to significant impacts", which seems to lead the conclusion of the study to meaninglessness. Rewrite it to avoid misunderstanding.

15. Line 669-670: "LULC changes that are likely to occur in the future can have negligible (2%) to significant impacts (20%) on the extreme flows of Mahanadi river basin." Again, I suggest not using words like "negligible" or "significant" when describing the percentage results.

---

## Author Comment (AC2) · 16 Feb 2021

Reviewer 2

We appreciate the comments and insights provided by Reviewer#2, and below in **bold** include our response to the comments.

**General Overview:**

The study investigates how the hydrological regime of the Mahanadi river basin would respond to the current and future land cover scenarios under a large-scale hydrological modelling framework. The recently released dataset LUH2, which has not yet extensively used in basin-scale hydrology, is firstly used in this study to provide current and future projected land cover scenarios. Although many studies have addressed the effects of future land cover changes on hydrological processes, I believe the novelty of the study lies with the consideration of parameter uncertainties of the physically based model VIC, which are widely used by the hydrology community when evaluating LULC and climate changes. Valuable insights are provided into the sensitivity of the model parameters to the model outputs and the interactions among the model parameters in producing changes of the hydrological regime within different LULC scenarios.

**We would like to thank the reviewer for this positive comment. We are happy to see the comment about the LUH2 dataset not yet being extensively used, and we hope to provide new insights on the hydrological processes of the basin using this dataset.**

To some point, I agree with RC1 that the parameter sensitivity tests are not properly designed and there lacks through analysis to link the sensitivity results with the LULC impacts. However, the authors agreed to re-conduct the sensitivity analysis and the calibration experiments with the involvement of more vegetation and soil-related parameters. I am looking forward to the new outcomes and modifications.

**Thanks for pointing this out. We realize this was the shortcoming in our study as highlighted by Reviewer 1. We are already making substantial changes in the model sensitivity, calibration methods, and the overall structure of the paper (as discussed in our reply to Reviewer 1) which will be presented in the revised version of the manuscript. Therefore, in this short response, it is important for the reviewer to recognize that we are focusing on the key comments raised by reviewer which are not directly related to the formatting or presentation of the paper (i.e., typos and unclear sentences) as the revised version will likely look significantly different.**

**Specific comments:**

1. Figure 2, please indicate in the figure title the time period of the LULC map and LAI data, and show the land use types in the legends rather than using abbreviations.

**We thank the reviewer for pointing this out. We will add the time period for both LULC map and LAI data, and will consider revising the legends providing we reach a good balance between required text and information from the figure. Notice that the land cover types shown in the legend are standard and widely used across the scientific community following IGBP convention.**

2. Line 198, explain what the exact meaning and the geographic scope of the "3 root zones".

**As part of your initial comment (and also from Reviewer 1), we are now redesigning the root zone allocation in our revised methodology (see our Response to Reviewer1 'Reply on RC1') and will add necessary explanations about this in the revised manuscript.**

3. Line 251, should "Klein-Gupta Efficiency" be "Kling-Gupta Efficiency (KGE)"? It is better if the equations of NSE, lnNSE and KGE could be given.

**Thanks for the correction/suggestion. As part of your initial comment (and also from Reviewer 1), we are now restructuring the manuscript, including its analysis. In order to simplify our analysis and to help with our discussion, we are now only using the Kling-Gupta Efficiency (KGE) metric. Note the KGE metric balances the contribution to the error coming from all three main components, namely correlation (e.g., timing/dynamics), variability (e.g., seasonality), and systematic bias, and is now a widely used metric in hydrometeorological studies. Some of the shortcomings of using NSE and benefits of using KGE are described in the Gupta et al. (2009). The revised version of the manuscript will include the KGE equation.**

4. Line 330-331, "In all the six cases of model run, meteorological forcing is held constant:: :", are the forcing data from the baseline scenario (2005) are used? Please give detail explanations.

**We realized this sentence can be confusing and affect the interpretation of our results. The meteorological data used is for the period 1990-2010. This period is used at all experiments presented in the paper. While the meteorological data used remain unchanged, we do however modify the land cover data to identify the hydrological impacts under LULC changes. These are represented by four distinct LULC maps of year 2005, 2015, 2050 and 2100 and two hypothetical scenarios with full land cover for Cropland and Forest, respectively. We name those LULC scenarios as baseline (2005), Present (2015), Near Future (2050), Far Future (2100) and the two hypothetical scenarios for cropland and forest. We will make sure this information is clearly presented in the revised version of the manuscript.**

5. Line 421, the time period 1990-2010 is used for analysing the effects of different LULC scenarios on the streamflow in the study. However, this covers the calibration and validation periods used for generating the VIC model parameters. In normal case, data used for calibrating models should not be used again for further analysis.

**We thank the reviewer for pointing this out. The separation between calibration and validation period we took in the initial part of the paper was to identify any possible significant deviations from the calibrated model in simulating key hydrological processes outside the calibration period. Our results indicated the expected reduction of performance but nothing to cause alarm and invalidate the experiment. For the revised version of the manuscript, the LULC scenarios will be analysed using only the common validation period (2000-2010) as pointed out by the reviewer.**

6. Line 503-520: this paragraph describes how the parameters influences the change of the streamflow as well as other water balance components. However, these descriptions seem to be deductive from the physical definitions of the parameters, rather than being concluded

from Fig 8. The results showed in Fig 8 are not well analysed and the sensitivity of parameters should be given in a more quantitative way.

**We thank the reviewer for raising this issue. We agree with the reviewer and as we are restructuring the revised version of the paper, we will take this comment into account to improve the discussion of the results.**